# The incidence rate and influence factors of hemolysis, lipemia, icterus in fasting serum biochemistry specimens

Gang Tian[1]*, Yu Wu[1], Xinrui Jin[1], Zhangrui Zeng[1], Xiujuan Gu[1], Tao Li[2], Xiu Chen[3], Guangrong Li[1], Jinbo Liu[1]*

1 Department of Clinical Laboratory Medicine, the Affiliated Hospital of Southwest Medical University, Luzhou, Sichuan, China, 2 Network Manage Center, the Affiliated Hospital of Southwest Medical University, Luzhou, Sichuan, China, 3 Department of Neurology, the Affiliated Hospital of Southwest Medical University, Luzhou, Sichuan, China

☯ These authors contributed equally to this work.
* tiangang@swmu.edu.cn (GT); liulab2019@163.com (JL)

**Data Availability Statement:** All relevant data are within the paper and its Supporting information files.

## Abstract

### Objective

Hemolysis, icterus, and lipemia (HIL) of blood samples have been a concern in hospitals because they reflect pre-analytical processes' quality control. However, very few studies investigate the influence of patients' gender, age, and department, as well as sample-related turnaround time, on the incidence rate of HIL in fasting serum biochemistry specimens.

### Methods

A retrospective, descriptive study was conducted to investigate the incidence rate of HIL based on the HIL index in 501,612 fasting serum biochemistry specimens from January 2017 to May 2018 in a tertiary university hospital with 4,200 beds in Sichuan, southwest China. A subgroup analysis was conducted to evaluate the differences in the HIL incidence rate by gender, age and department of patients, and turnaround time of specimens.

### Results

The incidence rate of hemolysis, lipemia and icterus was 384, 53, and 612 per 10,000 specimens. The male patients had a significantly elevated incidence of hemolysis (4.13% vs. 3.54%), lipemia (0.67% vs. 0.38%), and icterus (6.95% vs. 5.43%) than female patients. Hemolysis, lipemia, and icterus incidence rate were significantly associated with the male sex with an odds ratio (OR) of 1.174 [95% confidence interval (CI), 1.140–1.208], 1.757 (95%CI: 1.623–1.903), and 1.303 (95%CI: 1.273–1.333), respectively, (P<0.05). The hospitalized patients had a higher incidence of hemolysis (4.03% vs. 3.54%), lipemia (0.63% vs. 0.36%), and icterus (7.10% vs. 4.75%) than outpatients (P<0.001). Specimens with relatively longer transfer time and/or detection time had a higher HIL incidence (P<0.001). The Pediatrics had the highest incidence of hemolysis (16.2%) with an adjusted OR (AOR) of

**Funding:** This work was supported by the doctor of medicine start-up fund of the Affiliated Hospital of Southwest Medical University (18057), and the Luzhou-Southwest Medical University applied basic research project (2018LZXNYD-ZK30), and the Department of Science and Technology of Sichuan Province (2019YFH0010).

**Competing interests:** The authors have declared that no competing interests exist.

4.93 (95%*CI*, 4.59–5.29, *P*<0.001). The Neonatology department had the highest icterus incidence (30.1%) with an AOR of 4.93 (95%*CI*: 4.59–5.29, *P*<0.001). The Neonatology department (2.32%) and Gastrointestinal Surgery (2.05%) had the highest lipemia incidence, with an AOR of 1.17 (95%*CI*: 0.91–1.51) and 4.76 (95%*CI*: 4.70–5.53), both *P*-value <0.001. There was an increasing tendency of hemolysis and icterus incidence for children under one year or adults aged more than 40.

## Conclusion

Evaluation of HIL incidence rate and HIL-related influence factors in fasting serum biochemistry specimens are impartment to interpret the results more accurately and provide better clinical services to patients.

## 1. Introduction

Biochemical analysis of blood samples provides an essential basis for auxiliary clinical diagnosis and treatment decision-making. Clinical laboratory specimens sent to the laboratory often lead to rejection due to different reasons (e.g., hemolysis, clotted, insufficient volume, and lipemic specimens) [1]. Hemolysis, icterus, and lipemia (HIL) are common pre-analytical problems affecting the results of routine clinical tests, and interfere with the accurate measurement of various analytes, and may lead to wrong interpretations [2, 3]. In vitro hemolysis is the most prevalent pre-analytical error [4], and the proportion of hemolyzed samples received at the laboratory has been reported as high as 3.3% of all routine blood samples [5], accounting for 40% to 70% of all unsuitable samples identified [6]. The total incidence rate of lipemia in blood samples ranges from 0.5% to 2.5%, depending on the types of hospital and the proportion of inpatient and outpatient samples [7]. On the contrary, jaundice often occurs in neonates, with 4.5 per 100 person-hours of the overall incidence rate [8]. Therefore, identifying specimens with HIL is crucial for laboratories to reduce or eliminate these analytical interferents.

In recent years, sophisticated chemical analyzers have automatically detected the HIL status and reported the HIL index to evaluate their effect on routine analytes because of its integrated, automated use on the chemistry and immunochemistry platforms [9]. The HIL status and comments provide a fast and accurate way to determine HIL effects on each analyte, especially when each analyte's HIL index value is above the corresponding HIL alert index. Therefore, laboratory application of HIL status verifying the clinically significant HIL index is beneficial for reducing the laboratory turnaround time, re-tests, and specimen recollection [9]. Additionally, the Laboratory Information System (LIS) provides a helpful tool for monitoring and consulting based on comprehensive online data. Specimens with the wrong test items, sample types, containers can be identified and recorded on the LIS [10]. Therefore, the inclusion of the LIS and the HIL index in biochemical assays provides the means for evaluating analytic test results at the time of the whole process to decide whether they are reliable enough to be released to the requesting clinicians.

The study aimed to evaluate the incidence rate of HIL in an International Standardization Organization (ISO) 15189 accredited clinical laboratory in a tertiary university hospital; investigate whether the incidence rate of HIL is associated with gender, age, and department of patients, and sample turnaround time for providing adequate quality control and continuous improvement in the three fields of pre-analytical, analytical, and post-analytical processes.

## 2. Material and methods

### 2.1 Data sources

All clinical chemistry analyses were conducted in an ISO 15189 accredited clinical laboratory at the Department of Laboratory Medicine, Affiliated Hospital of Southwest Medical University, in Sichuan province, Southwest China. The electronic health record is currently DHC (DHC Software, Co, Ltd, China) which contains historical data covering the entire retrospective analysis period. Clinical data were collected from the electronic medical records at the time of specimen collection, including gender, age, diagnosis, departments of each patient, transfer time, detection time of specimens, specimen types, and the HIL index. This study was carried out by the Code of Ethics of the World Medical Association (Declaration of Helsinki). The data in this study is from the core laboratory clinical chemistry section and was collected as part of a study approved by the Affiliated Hospital of Southwest Medical University (No. 20180306089). Five hundred twenty-one thousand four hundred three serum biochemical specimens, including proteins, enzymes, metabolites, fasting, and postprandial glucose, blood lipids, and electrolytes, were collected from 36 clinical departments in the Affiliated Hospital of Southwest Medical University from January 1, 2017, to May 31, 2018. In order to decrease the influence of diet, we included specimens of fasting blood samples by searching the LIS records retrospectively. Data were further excluded if the sampling time was beyond the fasting blood collection period (6:00 am to 11:30 am). Finally, the authors and seven medical college students checked records to exclude potential postprandial specimens, especially postprandial glucose, pancreatitis markers related specimens (e.g., amylase, lipase, and amylopsin), and specimens of emergency because of the ingestion or not unknown, and plasma specimens. Data of 19,791 serum or plasma biochemical specimens were excluded because of the postprandial serum biochemical specimens or data without the HIL index or the transfer time of specimen > 12 hours. We finally included 501,612 eligible fasting serum biochemical specimens for analysis.

### 2.2 Laboratory confirmation of HIL index values

The HIL index was measured automatically by the ADVIA-2400 system (Siemens Healthcare Diagnostics). The HIL index were measured automatically by the ADVIA-2400 system (Siemens Healthcare Diagnostics). The serum HIL index feature of the ADVIA-2400 chemistry system can detect and produce a qualitative estimate absorbance value of three sets of wavelengths: hemolysis ($\lambda_1$ = 571 nm, $\lambda_2$ = 596 nm), lipemia ($\lambda_1$ = 658 nm, $\lambda_2$ = 694 nm), and icterus ($\lambda_1$ = 478 nm, $\lambda_2$ = 505 nm) [11]. The concentration ranges of HIL index value were set as follows: hemoglobin, < 45 mg/dL (-), 45–140 mg/dL (+), 140–235 mg/dL (++), 235–445 mg/dL (+++) and > 445 mg/dL (++++); lipemia, < 120 mg/dL (-), 120~245 mg/dL (+), 245~495 mg/dL (++), 495–995 mg/dL (+++) and > mg/dL 995 (++++); icterus, < 1.60 mg/dL (-), 1.60–6.50 mg/dL (+), 6.50–15.0 mg/dL (++), 15.0–28.0 mg/dL (+++) and > 28.0 mg/dL (++++). Clinical and Laboratory Standards Institute (CLSI) document C56A guides the use of serum indices to measure HIL interference and recommends the selection of assay-specific HIL cutoffs, above which HIL interferences will affect results. The HIL index in the study is defined as the lowest concentrations of HIL that interfere with chemical analyses, yielding a bias >10%.

### 2.3 Statistical analyses

Variables with a normal distribution were presented as mean and standard deviation (SD); otherwise, the median and interquartile range (IQR) was used. Wilcoxon rank-sum tests were applied to continuous variables, chi-square tests, and Fisher's exact tests were used for

categorical variables as appropriate. The univariate logistic regression analysis was used to evaluate the odds ratio (OR) of HIL, and variables with *P*-value < 0.10 were included in the multivariate logistic regression model to provide an adjusted OR (AOR) by gender, age, transfer time, detection time, hemolysis, lipemia, and icterus. Statistical significance was determined using two-tailed tests, and a value of $P < 0.05$ was accepted as the statistical significance limit. The data were analyzed using the SPSS software version 22 for Windows (SPSS, Chicago, IL, USA).

## 3. Results

### 3.1 Demographic and clinical data information

Data of 501,612 fasting serum biochemical specimens were included in the study, which contains 249,581 (49.8%) specimens of males and 252,031 (50.2%) specimens of females with a mean age of 52.0 (39.0–64.0) and 50.0 (36.0–63.0), respectively. In total, the incidence of hemolysis, lipemia, and icterus was 384, 53, and 612 per 10,000 specimens. The highest incidence rate of hemolysis (11.8%), lipemia (1.18%), and icterus (11.1%) occurred in young children aged zero to three years. With the increase of age, there was a decreased HIL incidence for patients aged four to 30 years and an increased HIL incidence when the age was more than 30 years. The males have higher incidence rate of hemolysis (4.13% vs. 3.54%, *P*<0.001), lipemia (0.67% vs. 0.38%, *P*<0.001) and icterus (6.96% vs. 5.43%, *P*<0.001) than those in females. There were also significantly longer transference and detection times for HIL specimens than non-HIL specimens (*P*<0.001). Compared with the outpatients, there was a higher incidence of hemolysis (4.03% vs. 3.54%, *P*<0.001), lipemia (0.63% vs. 0.36%, *P*<0.001), and icterus (7.10% vs. 4.75%, *P*<0.001) for specimens from the hospitalized patients. The HIL incidence was related to the department of specimens with 2.50%–14.3% hemolysis, 0.20%–1.18% lipemia, and 4.18%–16.7% icterus. The department of Pediatrics and Neonatology (14.3%), Gynecology & Obstetrics (1.18%), and the Department of Infectious diseases (16.7%) had the highest hemolysis, icterus, and lipemia incidence, respectively (Table 1).

### 3.2 Incidence of hemolysis in the top 10 clinical departments

Table 2 shows the hemolysis incidence in the top 10 clinical departments and the related clinical and laboratory characterizations. The department of Pediatrics had the highest incidence rate of hemolysis (16.2%) than neonatology (11.3%), Cardiology (6.74%), Plastic and Burn (6.62%), Neurology (6.53%), Digestive Diseases (4.76%), Endocrinology (4.19%), Respiratory Medicine (4.16%), Otolaryngology (3.43%), Nephrology (3.12%), and the other 26 clinical departments. The department of Pediatrics had the highest OR of hemolysis [5.06, 95% confidence interval (*CI*): 4.72–5.43)] and a decreased AOR of hemolysis (4.93, 95% *CI*: 4.59–5.29) by gender, age, transfer time and detection time, lipemia and icterus (Table 3). Compared with the female patients, the males had a significantly elevated incidence of hemolysis (4.13% vs. 3.54%), with a pooled OR of 1.174 (95%*CI*, 1.140–1.208, *P*<0.001) based on the top ten high hemolysis incidence departments (Fig 1).

### 3.3 The incidence rate of lipemia in the top 10 clinical departments

Table 4 shows the incidence rate of lipemia in the top 10 clinical departments. The department of neonatology has the highest incidence of lipemia (2.32%), which was higher than that in Gastrointestinal Surgery (2.05%), Neurosurgery (1.03%), Digestive Diseases (0.91%), Pediatrics (0.79%), Hepatobiliary Surgery (0.69%), Neurology (0.67%), Cardiothoracic Surgery (0.63%), Vascular Surgery (0.60%), Emergency (0.53%), and the other 26 clinical departments.

**Table 1. Patients' basic information and the incidence rate of hemolysis, lipemia, and icterus in 501,612 fasting serum biochemical specimens.**

| Clinical characteristics | Specimens (n = 501,612) | Hemolysis (yes = 19,252) (no = 482,360) | P value | Lipemia (yes = 2,634) (no = 498,978) | P value | Icterus (yes = 31,059) (no = 470,553) | P value |
|---|---|---|---|---|---|---|---|
| Total incidence rate | - | 19,252 (3.84%) | - | 0.53% | - | 6.12% | - |
| Age (y), Median (range) | 52.0 (39.0–64.0) [m] | 52.0 (38.0–65.0) [y] | | 49.0 (38.0–61.0) [y] | | 51.0 (39.0–63.0) [y] | |
| | 50.0 (36.0–63.0) [f] | 50.0 (37.0–63.0) [n] | | 51.0 (37.0–64.0) [n] | | 51.0 (37.0–64.0) [n] | |
| 0–3 | 13,896 (2.77%) | 1,633 (11.8%) | <0.001 | 164 (1.18%) | <0.001 | 1,545 (11.1%) | <0.001 |
| 4–14 | 43,210 (8.61%) | 1,583 (3.66%) | | 30 (0.07%) | | 378 (0.87%) | |
| 15–30 | 63,235 (12.6%) | 1,640 (2.59%) | | 235 (0.37%) | | 3,230 (5.10%) | |
| 31–45 | 94,133 (18.8%) | 2,879 (3.06%) | | 627 (0.67%) | | 6,136 (6.52%) | |
| 46–60 | 143,790 (28.7%) | 5,300 (3.69%) | | 911 (0.63%) | | 10,169 (7.07%) | |
| 61–80 | 127,325 (25.4%) | 5,501 (4.32%) | | 564 (0.44%) | | 8,608 (6.76%) | |
| ≥81 | 16,023 (3.19%) | 716 (4.47%) | | 103 (0.64%) | | 993 (6.19%) | |
| **Gender** | | | | | | | |
| Male | 249,581 (49.8%) | 10,319 (4.13%) | <0.001 | 1,671 (0.67%) | <0.001 | 17,371 (6.96%) | <0.001 |
| Female | 252,031 (50.2%) | 8,933 (3.54%) | | 963 (0.38%) | | 13,688 (5.43%) | |
| **Transfer time (h)** | 1.34 (0.64–4.13) | 1.48 (0.78–3.34) [y] | <0.001 | 1.38 (0.69–3.38) [y] | <0.001 | 1.38 (0.72–2.97) [y] | <0.001 |
| | | 1.34 (0.64–4.19) [n] | | 1.34 (0.64–4.14) [n] | | 1.34 (0.64–4.47) [n] | |
| | 1.01 (0.55–2.13) [i] | 0.92 (0.49–2.39) [i] | <0.001 | 0.99 (0.58–2.24) [i] | <0.001 | 1.04 (0.48–2.32) [i] | <0.001 |
| | 1.64 (0.64–3.56) [o] | 1.52 (0.79–3.47) [o] | | 1.61 (0.77–3.72) [o] | | 1.68 (0.80–3.99) [o] | |
| **Detection time (h)** | 1.00 (0.87–1.76) | 1.01 (0.87–1.76) [y] | <0.001 | 1.00 (0.87–1.83) [y] | <0.001 | 1.00 (0.88–1.79) [y] | <0.001 |
| | | 1.00 (0.86–1.73) [n] | | 1.00 (0.87–1.76) [n] | | 0.99 (0.77–1.30) [n] | |
| **Source of specimen** | | | | | | | |
| Hospitalized patients | 306,387 (61.1%) | 12,341 (4.03%) | <0.001 | 1,940 (0.63%) | <0.001 | 21,784 (7.10%) | <0.001 |
| Outpatients | 195,225 (38.9%) | 6,911 (3.54%) | | 694 (0.36%) | | 9,275 (4.75%) | |
| **Department of specimen** | | | | | | | |
| Internal Medicine | 128,352 (25.6%) | 6,431 (5.01%) | | 576 (0.45%) | | 7,088 (5.52%) | |
| Surgery | 92,236 (18.4%) | 2,308 (2.50%) | | 627 (0.68%) | | 5,622 (6.10%) | |
| Gynecology & Obstetrics | 11,389 (2.27%) | 335 (2.94%) | | 134 (1.18%) | | 476 (4.18%) | |
| Pediatrics and Neonatology | 9,704 (1.93%) | 1,392 (14.3%) | | 41 (0.42%) | | 1,355 (14.0%) | |
| Infectious Diseases | 9,658 (1.93%) | 263 (2.72%) | | 19 (0.20%) | | 1,613 (16.7%) | |
| Others | 250,273 (49.9%) | 8,523 (3.41%) | | 1237 (0.49%) | | 14,905 (5.96%) | |

Notes: m = male, f = female, y = yes, n = no, i = inpatients, o = outpatients.

The Neonatology department has the highest OR of lipemia (4.62, 95%*CI*: 3.72–5.73, *P*<0.001) in 36 clinical departments, whereas the AOR of lipemia was only 1.17 (95%*CI*: 0.91–1.51, *P* = 0.217) (Table 5). The department of Gastrointestinal Surgery has a second higher OR (4.22, 95%*CI*: 3.66–4.89, *P*<0.001), and the highest AOR (4.76, 95%*CI*: 4.70–5.53, *P*<0.001) than that in other departments. Besides, the males have a significantly elevated incidence of lipemia (0.67% vs. 0.38%) in comparison with the female patients, with a pooled OR of 1.757 (95%*CI*: 1.623–1.903, *P*<0.001) based on the top ten high lipemia incidence departments (Fig 2).

### 3.4 The incidence rate of icterus in the top 10 clinical departments

Table 6 shows the incidence rate of icterus in the top 10 clinical departments. The Neonatology Department has the highest incidence rate of icterus (30.0%) than Infectious Diseases

**Table 2. Incidence of hemolysis in the top 10 clinical departments (n = 127,155).**

| Department | incidence | Male | Female | Age | Transfer time (h) | Detection time (h) |
|---|---|---|---|---|---|---|
| | %(n) | % (n) | % (n) | (y/d*) | | |
| Pediatrics (5,953) | 967 (16.2) | 590/3,523 (16.8) | 377/2,430 (15.5) | 1.0 (0–4.0) | 1.0 (0.6–1.6) | 1.0 (0.8–1.4) |
| Neonatology (3,751) | 425 (11.3) | 250/2,267 (11.0) | 175/1,484 (11.8) | 7.3 (4.5–13.8) * | 1.4 (1.0–1.9) | 1.4 (1.0–1.9) |
| Cardiology (31,774) | 2,141 (6.74) | 1,293/1,8047 (7.16) | 848/13,727 (6.18) | 64.4±14.2 | 5.7 (1.9–10.5) | 1.1 (1.0–2.0) |
| Plastic and burn (3,263) | 216 (6.62) | 141/2,128 (6.63) | 75/1,135 (6.61) | 5.0 (2.0–42.8) | 1.9 (1.0–2.6) | 1.4 (1.0–2.2) |
| Neurology (20,807) | 1,358 (6.53) | 852/11,439 (7.45) | 506/9,368 (5.40) | 61.2±16.0 | 2.3 (1.5–2.9) | 1.2 (1.0–2.1) |
| Digestive Diseases (15,980) | 761 (4.76) | 435/9,041 (4.81) | 326/6,939 (4.70) | 57.2±15.1 | 1.9 (1.1–3.8) | 1.0 (0.8–1.7) |
| Endocrinology (13,519) | 566 (4.19) | 258/5,817 (4.44) | 308/7,702 (4.01) | 54.1±16.8 | 3.0 (1.4–14.1) | 1.0 (0.9–1.8) |
| Respiratory Medicine (17,315) | 721 (4.16) | 512/11,935 (4.29) | 209/5,380 (3.88) | 64.2±14.9 | 2.7 (1.0–7.1) | 1.1 (0.9–1.9) |
| Otolaryngology (5,600) | 192 (3.43) | 125/3,506 (3.57) | 67/2,094 (3.20) | 46.2±19.8 | 3.8 (1.1–7.8) | 1.3 (1.0–2.2) |
| Nephrology (9,193) | 287 (3.12) | 176/5,044 (3.49) | 111/4,149 (2.68) | 51.8±18.1 | 11.0 (1.8–14.4) | (0.9–1.9) |

Notes: y = yes; d = day; h = hour.

Department (16.7%), Hepatobiliary Surgery (13.6%), Digestive Diseases Department (10.2%), Cardiology (6.96%), Emergency Department (6.81%), Vascular Surgery (6.36%), Cardiothoracic Surgery (6.18%), Outpatient Service (5.98%), Gastroenterology (5.54%) and the other 26 clinical departments. The Neonatology Department has the highest OR of icterus (6.71, 95% *CI*: 6.26–7.21, *P*<0.001), and the AOR of 7.62 (95%*CI*: 7.04–8.24, *P*<0.001) (Table 7). Compared with the female patients, the males have a significantly elevated incidence of icterus (6.95% vs. 5.43%), with an OR of 1.303 (95%*CI*: 1.273–1.333, *P*<0.001) based on the top ten high icterus incidence departments (Fig 3).

### 3.5 The influence of age for hemolysis, lipemia, and icterus incidence

Among 501,612 fasting serum biochemistry specimens, HIL incidence varied according to the age group. There was a significantly decreased HIL incidence for patients aged from zero to one year. On the contrary, there was a relatively increased risk for hemolysis and lipemia (both age >18 years) and icterus (18< age < 60 years), with the highest cases of hemolysis and icterus aging from 40 to 70 years. Significantly, the HIL incidence slowly increased from 18 years and reached the highest platform for hemolysis (40–70 years), lipemia (30–60 years), and icterus (40–80 years), shown in Fig 4.

**Table 3. Logistic regression analysis of the OR and the AOR of hemolysis in the top ten departments (n = 127,155).**

| Departments (n) | OR | 95% CI | P | AOR | 95%CI | P |
|---|---|---|---|---|---|---|
| Pediatrics (5,953) | 5.06 | 4.72–5.43 | <0.001 | 4.93 | 4.59–5.29 | <0.001 |
| Neonatology (3,751) | 3.25 | 2.94–3.60 | <0.001 | 1.15 | 1.12–1.10 | <0.001 |
| Cardiology (31,774) | 1.91 | 1.83–2.00 | <0.001 | 1.15 | 1.11–1.18 | <0.001 |
| Plastic and burn (3,263) | 1.79 | 1.55–2.05 | <0.001 | 1.16 | 1.12–1.19 | <0.001 |
| Neurology (20,807) | 1.81 | 1.71–1.91 | <0.001 | 1.15 | 1.12–1.19 | <0.001 |
| Digestive Diseases (15,980) | 1.26 | 1.17–1.36 | <0.001 | 1.16 | 1.13–1.19 | <0.001 |
| Endocrinology (13,519) | 1.10 | 1.01–1.20 | 0.032 | 1.16 | 1.13–1.20 | <0.001 |
| Respiratory Medicine (17,315) | 1.09 | 1.01–1.78 | 0.023 | 1.16 | 1.12–1.19 | <0.001 |
| Otolaryngology (5,600) | 0.89 | 0.77–1.02 | 0.109 | 1.16 | 1.13–1.20 | <0.001 |
| Nephrology (9,193) | 0.80 | 0.72–0.91 | 0.000 | 1.16 | 1.13–1.20 | <0.001 |

Notes: AOR was adjusted by gender, age, transfer time, detection time, lipemia, and icterus.

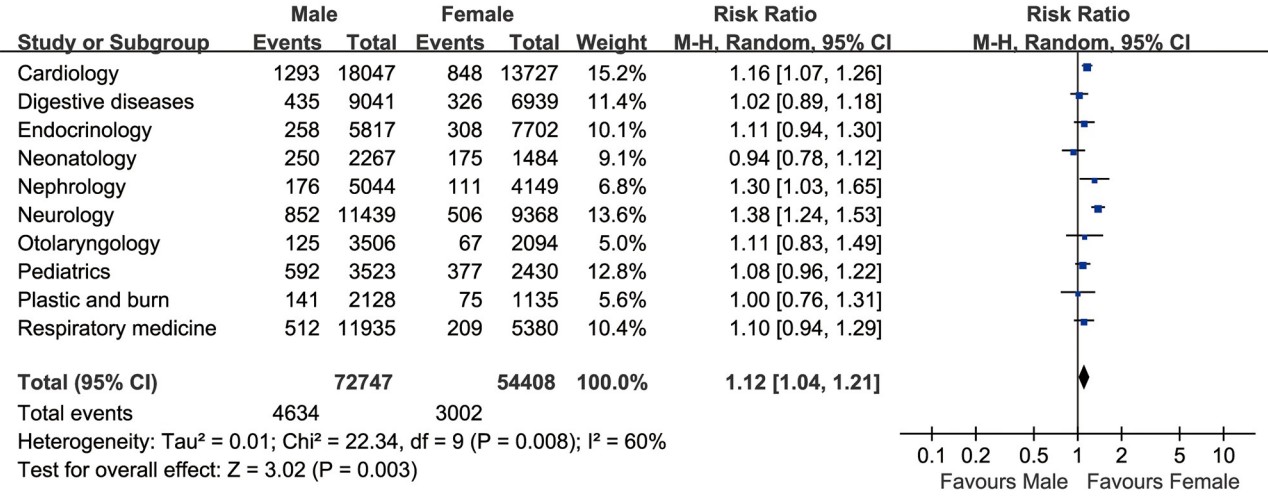

**Fig 1. The pooled OR of hemolysis based on the top ten high hemolysis incidence departments (n = 127,155).**

## 4. Discussion

Clinical laboratory tests play an integral role in medical decision-making, and therefore, the results must be reliable and accurate [12]. However, lipemia, icterus, or hemolysis are encountered routinely in clinical practice, affecting the results of routine clinical tests [13]. Although hemolysis can be avoided by careful drawing and handling the whole-blood samples, it is one of the significant risks during blood drawing and the most common pre-analytical error in the clinic [14, 15]. In this study, the total incidence rate of hemolysis is 3.84%, similar to Wan Azman et al. report (3.3%) [6]. Hemolysis can be caused by many pre-analytical causes associated with venipuncture, sample collection, transportation methods, temperature, sample handling, and delayed processing [16, 17]. During venipuncture, many conditions cause hemolysis, including forceful evacuation of a syringe into a tube, prolonged tourniquet application, vigorous mixing of the blood collected into the tube, and the use of inappropriate needles [18]. In this study, we found that Pediatrics had the highest incidence of hemolysis (16.2%) with an adjusted OR (AOR) of 4.93 (95%*CI*, 4.59–5.29, *P*<0.001). A previous study

**Table 4. The top 10 departments with the high incidence rate of lipemia (n = 103,857).**

| Departments (*n*) | IR | Male | Female | Age | Transfer time (h) | Detection time (h) |
|---|---|---|---|---|---|---|
| | (%) | (%) | (%) | (y/d*) | | |
| Neonatology (3,751) | 87 (2.32) | 49/2,267 (2.16) | 38/1,484 (2.56) | 7.0 (4.2–14.5) * | 1.3 (1.0–1.8) | 1.3 (0.9–2.0) |
| Gastrointestinal surgery (10,242) | 210 (2.05) | 109/6,000 (1.82) | 101/4,242 (2.38) | 58.3±14.4 | 2.3 (1.4–7.6) | 1.3 (1.0–2.1) |
| Neurosurgery (12,583) | 129 (1.03) | 80/7,173 (1.12) | 49/5,410 (0.91) | 53.8±15.8 | 3.8 (1.7–9.1) | 1.4 (1.0–2.1) |
| Digestive diseases (15,980) | 145 (0.91) | 83/9,041 (0.92) | 62/6,939 (0.89) | 51.3±16.1 | 1.6 (0.9–10.7) | 1.0 (0.9–1.8) |
| Pediatrics (5,953) | 47 (0.79) | 30/3,523 (0.85) | 17/2,430 (0.70) | 2.48±0.96 | 1.0 (0.6–1.5) | 0.9 (0.7–1.4) |
| Hepatobiliary surgery (11,135) | 770 (0.69) | 46/5,702 (0.81) | 31/5,433 (0.57) | 61.0±16.0 | 3.7 (1.1–10.5) | 1.7 (1.0–3.6) |
| Neurology (20,807) | 140 (0.67) | 80/11,439 (0.70) | 60/9,368 (0.64) | 64.0±17.9 | 3.8 (1.7–9.1) | 1.4 (1.0–2.1) |
| Cardiothoracic surgery (9,552) | 60 (0.63) | 33/5,699 (0.58) | 27/3,853 (0.70) | 54.7±18.8 | 2.9 (1.4–3.4) | 1.2 (1.0–1.9) |
| Vascular surgery (6,837) | 142 (0.60) | 27/2,957 (0.91) | 15/3,880 (0.39) | 55.9±12.6 | 3.5 (1.1–10.4) | 1.4 (1.0–2.0) |
| Emergency (7,017) | 37 (0.53) | 27/4,506 (0.61) | 10/2511 (0.40) | 45.9±15.5 | 3.3 (0.8–8.0) | 1.2 (1.0–2.1) |

Note: y = yes; d = day; h = hour.

**Table 5. Logistic regression analysis of the OR and the AOR of lipemia in the top ten departments (n = 103,857).**

| Departments (n) | OR | 95% CI | P | AOR | 95%CI | P |
|---|---|---|---|---|---|---|
| Neonatology (3,751) | 4.62 | 3.72–5.73 | <0.001 | 1.17 | 0.91–1.51 | 0.217 |
| Gastrointestinal Surgery (10,242) | 4.22 | 3.66–4.89 | <0.001 | 4.76 | 4.70–5.53 | <0.001 |
| Neurosurgery (12,583) | 2.01 | 1.68–2.40 | <0.001 | 2.32 | 1.93–2.78 | <0.001 |
| Digestive diseases (15,980) | 1.78 | 1.50–2.10 | <0.001 | 1.42 | 1.20–1.69 | <0.001 |
| Pediatrics (5,953) | 1.52 | 1.14–2.03 | 0.005 | 0.92 | 0.68–1.26 | 0.610 |
| Hepatobiliary surgery (11,135) | 1.33 | 1.06–1.67 | 0.014 | 1.05 | 0.83–1.32 | 0.694 |
| Neurology (20,807) | 1.30 | 1.10–1.54 | 0.003 | 1.26 | 1.06–1.50 | 0.011 |
| Cardiothoracic surgery (9,552) | 1.20 | 0.93–1.55 | 0.160 | 1.27 | 0.98–1.65 | 0.071 |
| Vascular surgery (6,837) | 1.17 | 0.86–1.59 | 0.305 | 1.30 | 0.96–1.78 | 0.094 |
| Emergency (7,017) | 1.00 | 0.73–1.39 | 0.980 | 0.95 | 0.68–1.32 | 0.751 |

Note: AOR was adjusted by gender, age, transfer time, detection time, hemolysis, and icterus.

found that newborns and children typically have more limited venous access and much smaller veins, causing an approximately 13% hemolysis rate for all blood draws in these groups [19]. These reasons may explain the high incidence rate of hemolysis, 16.2%, and the increasing tendency in the Pediatrics department for young children under one year old. As nurses from clinical departments have participated in standardized sampling, transferring training during ISO 15189 accreditation, the other top 10 high hemolysis rates departments have a similar incidence rate of hemolysis except for pediatrics and neonatology, with an AOR ranging from 1.15 to 1.16. Besides, we found a higher incidence of hemolysis for specimens from hospitalized patients than in outpatients (4.03% vs. 3.54%). Although it is hard to explain that higher hemolysis occurs in hospitalized patients compared to outpatients, enhancing nurses' training on venipuncture techniques is helpful to reduce the incidence of hemolysis. In addition, our study found longer transportation times for those hemolysis specimens than non-hemolysis specimens (1.48h vs. 1.34h, P<0.001), demonstrating that longer transit time increased the risk of hemolysis [20]. When in vitro hemolysis occurs, RBCs release intracellular contents (e.g., potassium ion and hemoglobin) into the serum or plasma, causing a significant laboratory

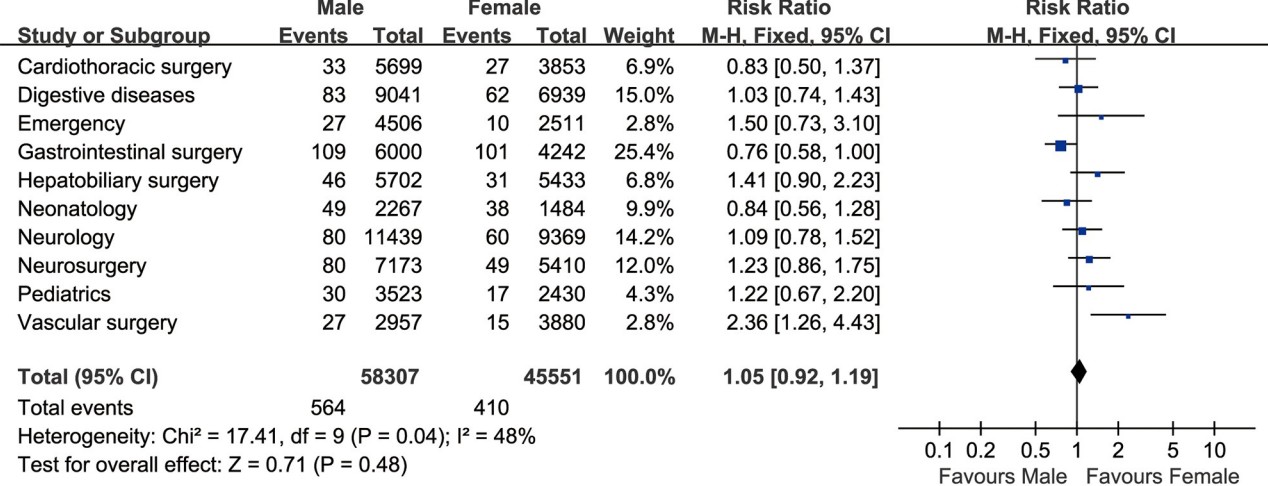

**Fig 2. The pooled OR based on the top ten high lipemia incidence departments (n = 103,857).**

**Table 6. The top 10 departments with the high incidence rate of icterus (n = 259,887).**

| Departments (n) | IR | Male | Female | Age | Transfer time (h) | Detection time (h) |
|---|---|---|---|---|---|---|
| | (%) | (%) | (%) | (y/d*) | | |
| Neonatology (3,751) | 1,127 (30.1) | 709/2,267 (31.3) | 418/1,484 (28.2) | 7.0 (4.2–14.5) * | 1.4 (0.9–2.1) | 1.0 (0.8–1.3) |
| Infectious Diseases (9,658) | 1,613 (16.7) | 1,281/6,933 (18.5) | 332/2,725 (12.2) | 48.8±14.0 | 3.6 (1.3–10.4) | 1.0 (0.9–1.8) |
| Hepatobiliary Surgery (11,135) | 1,518 (13.6) | 812/5,702 (14.3) | 706/5,433 (13.0) | 57.5±14.2 | 2.2 (1.2–3.4) | 0.9 (1.0–1.8) |
| Digestive Diseases (15,980) | 1,630 (10.2) | 1,033/9,041 (11.4) | 597/6,939 (8.60) | 55.9±14.9 | 1.9 (1.1–3.3) | 1.0 (0.8–1.6) |
| Cardiology (31,774) | 2,211 (6.96) | 1,248/18,047 (6.92) | 963/13,727 (7.02) | 65.1±13.7 | 2.4 (1.4–6.3) | 1.0 (0.9–1.5) |
| Emergency Pharmacy (7,017) | 478 (6.81) | 362/4,506 (8.03) | 116/2,511 (4.62) | 54.6±15.4 | 3.6 (1.2–15.6) | 1.0 (1.0–1.7) |
| Vascular Surgery (6,837) | 435 (6.36) | 206/2,957 (6.97) | 229/3,880 (5.90) | 54.0±16.4 | 2.6 (1.0–10.1) | 1.1 (0.9–2.0) |
| Cardiothoracic Surgery (9,552) | 590 (6.18) | 333/5,699 (5.84) | 257/3,853 (6.67) | 51.7±15.3 | 2.6 (1.4–3.3) | 1.0 (0.9–1.6) |
| Outpatient Service (15,3941) | 9,198 (5.98) | 4,815/67,975 (7.09) | 4,383/85,966 (5.1) | 46.2±15.6 | 0.6 (0.4–0.9) | 1.0 (0.7–1.0) |
| Gastrointestinal Surgery (10,242) | 567 (5.54) | 375/6,000 (6.25) | 4.53 (192/4,242) | 58.8±14.6 | 1.9 (1.1–3.3) | 1.0 (0.8–1.6) |

Note: y = yes; d = day; h = hour.

error source [21]. Consequently, it is crucial to focus on standardization of blood collection practices, staff training, and quickly transportation of blood specimens to reduce hemolysis rates in the pre-analytical process.

Lipemia is the most frequent endogenous interference that can influence various laboratory methods. The most common pre-analytical cause of lipemic samples is an inadequate blood sampling interval after the meal or parenteral administration of synthetic lipid emulsions [22]. We found that the incidence rate of lipemia was 0.53% after removing related influence factors. The Neonatology Department had the highest incidence of lipemia (2.32%) with an OR 4.62, whereas the AOR adjusted by gender, age, transfer time, detection time, hemolysis, and icterus was insignificant decreased (AOR = 1.17, 95%$CI$: 0.91–1.51, $P$ = 0.217). On the contrary, Gastrointestinal Surgery had the second higher incidence of lipemia (2.05%) with an OR of 4.22 and an increased AOR of 4.76 (95%$CI$: 4.70–5.53, $P<0.001$). Intravenous lipid emulsion infusion is a widely used way for total parenteral nutrition, an antidote for poisonings of local anesthetics, or the diluent for poorly water-soluble medications (e.g., propofol), and patients of all ages with feeding and gastrointestinal issues [22–24]. Blood collection in patients who have recently received intravenous lipid emulsion therapy, and drug therapy (e.g., glucocorticoids, antiretroviral medications, protease inhibitors, and non-selective beta-adrenergic antagonists),

**Table 7. Logistic regression analysis of the OR and the AOR of icterus in the top ten clinical departments (n = 259,887).**

| Departments (n) | OR | 95% CI | P | AOR | 95%CI | P |
|---|---|---|---|---|---|---|
| Neonatology (3,751) | 6.71 | 6.26–7.21 | <0.001 | 7.62 | 7.04–8.24 | <0.001 |
| Infectious Diseases (9,658) | 3.15 | 2.98–3.33 | <0.001 | 3.56 | 3.37–3.77 | <0.001 |
| Hepatobiliary Surgery (11,135) | 2.46 | 2.33–2.60 | <0.001 | 2.66 | 2.51–2.81 | <0.001 |
| Digestive Diseases (15,980) | 1.76 | 1.67–1.86 | <0.001 | 1.74 | 1.65–1.84 | <0.001 |
| Cardiology (31,774) | 1.14 | 1.09–1.20 | <0.001 | 1.24 | 1.18–1.30 | <0.001 |
| Emergency Pharmacy (7,017) | 1.11 | 1.01–1.22 | 0.030 | 1.30 | 1.18–1.43 | <0.001 |
| Vascular Surgery (6,837) | 1.03 | 0.93–1.14 | 0.556 | 1.09 | 0.99–1.20 | 0.083 |
| Cardiothoracic Surgery (9,552) | 0.98 | 0.92–1.09 | 0.951 | 0.97 | 0.89–1.06 | 0.481 |
| Outpatient Service (15,3941) | 0.96 | 0.94–0.98 | 0.001 | 0.73 | 0.71–0.75 | <0.001 |
| Gastrointestinal Surgery (10,242) | 0.89 | 0.81–0.97 | 0.005 | 0.81 | 0.80–0.82 | <0.001 |

Note: AOR was adjusted by gender, age, transfer time, detection time, hemolysis, and lipemia.

| Study or Subgroup | Male Events | Total | Female Events | Total | Weight | Odds Ratio M-H, Random, 95% CI |
|---|---|---|---|---|---|---|
| Cardiology | 1248 | 18047 | 963 | 13727 | 10.9% | 0.98 [0.90, 1.07] |
| Cardiothoracic surgery | 333 | 5699 | 257 | 3853 | 9.5% | 0.87 [0.73, 1.03] |
| Digestive diseases | 1033 | 9041 | 597 | 6939 | 10.6% | 1.37 [1.23, 1.52] |
| Emergency | 362 | 4506 | 116 | 2511 | 8.6% | 1.80 [1.45, 2.24] |
| Gastroenterology | 375 | 6000 | 192 | 4242 | 9.3% | 1.41 [1.18, 1.68] |
| Hepatobiliary surgery | 812 | 5702 | 706 | 5433 | 10.6% | 1.11 [1.00, 1.24] |
| Infectious diseases | 1281 | 6933 | 332 | 2725 | 10.2% | 1.63 [1.43, 1.86] |
| Neonatology | 709 | 2267 | 418 | 1484 | 10.0% | 1.16 [1.01, 1.34] |
| Outpatient service | 4815 | 67975 | 4383 | 85966 | 11.4% | 1.42 [1.36, 1.48] |
| Vascular surgery | 206 | 2957 | 229 | 3880 | 9.0% | 1.19 [0.98, 1.45] |
| | | | | | | |
| Total (95% CI) | | 129127 | | 130760 | 100.0% | 1.26 [1.11, 1.43] |
| Total events | 11174 | | 8193 | | | |

Heterogeneity: Tau² = 0.03; Chi² = 111.21, df = 9 (P < 0.00001); I² = 92%
Test for overall effect: Z = 3.67 (P = 0.0002)

**Fig 3. The pooled OR based on the top ten high icterus incidence departments (n = 259,887).**

can indirectly cause lipemia [25]. Additionally, increasing lipid concentrations have been associated with hemolysis frequency [26]. In this study, we randomly selected 901 lipemic serum samples from 1,022 lipemic patients in the Department of Gastrointestinal Surgery. We reviewed electronic medical records, and the results indicated that the proportion of patients who received parenteral nutrition solution (containing amino acids and fat emulsion) before or after surgery was 79.4% (715/901). Therefore, our data revealed that the most common causes of the markedly elevated lipemic index in Gastrointestinal Surgery were lipid emulsions before or after surgery. Currently, the management of lipemic specimens in the clinical

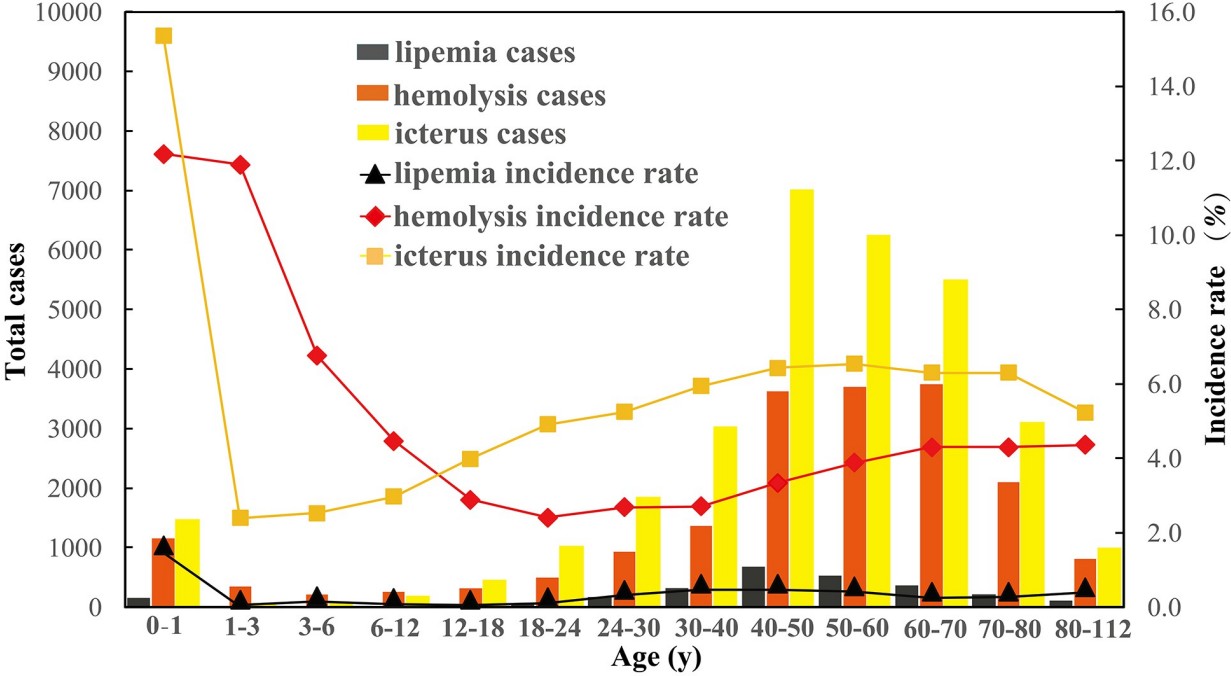

**Fig 4. The influence of age on HIL incidence (n = 501,612).**

laboratory is not standardized, and analytical methods for measuring lipemia are also heterogeneous [27]. Thus, the HIL index may provide more advice for further specimen preparation, especially the process of ultracentrifugation of lipid specimens to provide more accurate results to the clinical laboratory and benefit patients [28].

Icterus is a yellowish pigmentation of the skin and mucous membrane, mainly caused by an increased bilirubin level and its overproduction in the liver. Neonatal jaundice is a relatively prevalent disease in neonates, and more than half of newborns and 80% of preterm children develop clinical symptoms of jaundice [29]. Neonatal jaundice occurs mainly in premature infants, and the main clinical manifestations are yellow stains on the sclera, skin, and mucosa membranes of infants. This study found that the Neonatology department had the highest incidence of icterus (30.1%), similar to the previous report in Turkish, 31% [30]. Many pathogenic processes can cause jaundice, especially sepsis, hepatic necrosis, intrahepatic biliary obstruction, and alcoholic liver disease [31, 32]. Commonly, jaundice is a late event in the course of severe sepsis, whereas it can appear at an early stage of sepsis, even in the absence of fever or leukocytosis [33]. The incidence of jaundice was approximately 34% in septic patients, and the overall mortality rate of these patients was 61% [34]. Bile duct tumor thrombus is a significant cause of hepatocellular carcinoma complicated with obstructive jaundice, and its incidence is 1.2% to 9.0% [35]. This study found that the Infectious Diseases department and Hepatology Surgery department caused a high incidence of icterus, ranging from 13.6% to 16.7%, similar to previous studies.

Automated biochemical analyzers and reagents were recently used to minimize HIL interferences, whereas the results were not satisfying. Typically, a sample should be rejected when there is a high risk of reporting an unreliable result [9]. However, blood collection or re-correction is challenging in clinical practice, especially in newborns, young children, and other patients whose venous access is difficult (i.e., elderly or critically ill) [36, 37]. Therefore, educating doctors and nurses who collect the samples may minimize the frequency of unsuitable specimens. Besides, re-testing the specimens using another chemical analyzer or detection methods unaffected by the HIL may also minimize the hemolysis and icterus interference effects [38, 39].

This study has some limitations. Firstly, it is hard to ensure each specimen is fasting though we try our best to decrease the influence of diet for the measurements of routine serum biochemical specimens. Secondly, we did not assess the effect of hemolysis, lipemia, and jaundice on biochemical index results. Finally, we found that male patients had a significantly elevated incidence of hemolysis, lipemia, and icterus than female patients, whereas we cannot explain potential reasons and influence factors.

## 5. Conclusions

Our results indicated that the incidence of hemolysis, lipidemia and icterus was related to age, gender, transfer time, detection time, and patients' department. The HIL index is a valuable tool for evaluating HIL incidence in fasting serum biochemistry specimens. These findings are crucial for estimating the accuracy of results and optimizing the whole analytical process to provide adequate quality assurance in laboratory tests.

## Supporting information

**S1 Data.**
(XLSX)

## Author Contributions

**Conceptualization:** Gang Tian, Xiu Chen, Jinbo Liu.

**Data curation:** Yu Wu, Xinrui Jin, Xiujuan Gu, Tao Li.

**Formal analysis:** Zhangrui Zeng.

**Investigation:** Guangrong Li.

**Methodology:** Guangrong Li.

**Project administration:** Xiu Chen, Guangrong Li, Jinbo Liu.

**Resources:** Zhangrui Zeng.

**Software:** Zhangrui Zeng, Xiujuan Gu, Tao Li.

**Validation:** Yu Wu, Xinrui Jin, Xiujuan Gu, Tao Li.

**Visualization:** Xiu Chen, Jinbo Liu.

**Writing – original draft:** Gang Tian.

**Writing – review & editing:** Gang Tian.

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
