## [Decision Letter · Decision Letter 0]

6 Jul 2021

PONE-D-21-18605

The incidence rate of hemolysis, lipemia，icterus and influence factors: A single-center retrospective study of 501,612 fasting serum biochemistry specimens

PLOS ONE

Dear Dr. Tian,

Thank you for submitting your manuscript to PLOS ONE. After careful consideration, we feel that it has merit but does not fully meet PLOS ONE’s publication criteria as it currently stands. Therefore, we invite you to submit a revised version of the manuscript that addresses the points raised during the review process.

Specifically both reviewers expressed concerns over the lack of detail in the described methods and how the fasting information of the patients had been obtained. In addition,one reviewer thought a brief discussion of your results as related to the turnaround time for early diagnosis and treatment would also help improve the manuscript. Several other minor comments related to grammar were also listed.

We look forward to receiving your revised manuscript.

Kind regards,

Colin Johnson, Ph.D.

Academic Editor

PLOS ONE

Journal Requirements:

Reviewers' comments:

Reviewer's Responses to Questions

**Comments to the Author**

1. Is the manuscript technically sound, and do the data support the conclusions?

Reviewer #1: Yes

Reviewer #2: Partly

2. Has the statistical analysis been performed appropriately and rigorously? 

Reviewer #1: Yes

Reviewer #2: Yes

3. Have the authors made all data underlying the findings in their manuscript fully available?

Reviewer #1: Yes

Reviewer #2: Yes

4. Is the manuscript presented in an intelligible fashion and written in standard English?

Reviewer #1: Yes

Reviewer #2: Yes

5. Review Comments to the Author

Reviewer #1: Dear Authors,

Thank you for the interesting manuscript. My recommendations have been listed below for this manuscript:

1. Page 4: “The HIL index is defined as the lowest concentrations of HIL that interfere with chemical analyses, yielding a bias >10% based on the Clinical and Laboratory Standards Institute (CLSI) document C56-A (CLSI-C56-A).” In the CLSI document, an allowable bias of 10% is just presented as an example and cannot be generalized. Therefore, this sentence should be re-written.

2. Page 6: “We finally included 501,612 eligible fasting serum biochemical specimens for analysis.” Here, please explain how the fasting information of the patients had been obtained.

3. Page 17: “A previous study found that longer transit time increased the risk of hemolysis [27]. Meanwhile, it is crucial to reduce turnaround time for early diagnosis and treatment by providing the safety and pleasure of patients [26].” Please add own results and discuss them.

4. Page 17-18: “Besides, another study also indicated that in vivo aging of red blood cells is associated with increased cellular density, corresponding to increased cell age [28]. These findings may, in part, explain the increased tendency of hemolysis in adults aged more than 40.” In my opinion, here, the increasing cell age have been confused with the patient age. It should be reconsidered.

5. Page 19: “Our data revealed that the most common suspected causes of the markedly elevated lipemic index were lipid emulsion (Gastrointestinal Surgery) and hemolysis (Neonatology Department), consistent with these previous reports.” Here, please give information about the patients treated with lipid emulsion. Otherwise, this statement should be reconsider.

6. Page 20: “creatine” or “creatinine”?

7. The “hemolysis in vitro” should be changed to “in vitro hemolysis” thoroughly the manuscript.

8. Discussion have been written as if a book chapter. It should be shortened to the purpose of the study.

9. The style of the references should be changed according to the requirements of the journal.

Reviewer #2: The incidence rate of hemolysis, lipemia，icterus and influence factors: A single-center retrospective study of 501,612 fasting serum biochemistry specimens

Comment: The title should be revised, there is no need to indicate the sample size. Influencing factors may not fit based on the reasons that authors will find in major comments.

Major revisions: About Study design:

Authors should emphasize that this a descriptive study in their methodology . The authors indicated that the objective was to evaluate and link pre analytical factors like diet, drugs, collecting, handling, transportation, and preparing specimens. Yet, in their results, they are not interpreted and related to the HIL. Again, there are no methodology that highlight or show how these factors were collected.

• So, authors should refine their topic as well as their objective and as well as their research question

• Authors should refine their study and emphasize on gender and department according to their results presentation and these the tangible data collected according to their retrospective studies

• More of the findings, should be discussed generally and maybe recommended

In discussion part:

Authors should discuss specifically on the effects of the HIL for the lab results according to each parameter. For eg. physiological lipemia may not a wide number of tests being requested. Hemolysis may affect only hematology and maybe Potassium ion results but not others.

In conclusion part: Authors said they studied influence of factors of HIL, I suggest they should remove these factors as they were not investigated, rather they assumed theoretically as seen in their discussion.

Authors should rather emphasize on incidence rate as the topic highlights.

6. PLOS authors have the option to publish the peer review history of their article (what does this mean?). If published, this will include your full peer review and any attached files.

Reviewer #1: **Yes: **Şerif Ercan

Reviewer #2: **Yes: **Jean Baptiste Niyibizi

---

## [Author Response · Author response to Decision Letter 0]

15 Dec 2021

Reviewer #1: 

Dear Authors,

Thank you for the interesting manuscript. My recommendations have been listed below for this manuscript:

1. Page 4: "The HIL index is defined as the lowest concentrations of HIL that interfere with chemical analyses, yielding a bias >10% based on the Clinical and Laboratory Standards Institute (CLSI) document C56-A (CLSI-C56-A)." In the CLSI document, an allowable bias of 10% is just presented as an example and cannot be generalized. Therefore, this sentence should be rewritten.

Response: Thank you for spending your valuable time reading the manuscript and giving insightful suggestions to help us improve the quality of our manuscript. According to your suggestion, we carefully checked this sentence and rewrote it in the revised manuscript.

2. Page 6: "We finally included 501,612 eligible fasting serum biochemical specimens for analysis." Here, please explain how the fasting information of the patients had been obtained.

Response: Thanks for these insightful suggestions. The study investigated the incidence rate of hemolysis, lipemia, icterus, and influence factors based on fasting serum biochemistry specimens. Therefore, we used SIEMENS ADVIA 2400 biochemical analyzers to perform routine biochemical index tests, including proteins, enzymes, metabolites, fasting and postprandial glucose, blood lipids, and electrolytes. In order to decrease the influence of diet, we only included specimens labeled fasting blood samples (verification by blood collection nurses) by searching the LIS records retrospectively. Besides, HIL data were further excluded if the sampling time was beyond the fasting blood collection period (6:00 am to 11:30 am). Finally, the authors and seven medical college students separately checked each record to exclude potential postprandial specimens, especially postprandial glucose, pancreatitis markers (e.g., amylase, lipase, and amylopsin), and plasma specimens. In general, we took more than eleven months to check patients' data and information, including gender, age, departments transportation time, and the detection time of specimens. Though we try our best to decrease the influence of diet for the measurements of routine serum biochemical specimens, it is hard to ensure each specimen is fasting. Therefore, we added these limitations in the discussion of the revised manuscript. 

3. Page 17: "A previous study found that longer transit time increased the risk of hemolysis [27]. Meanwhile, it is crucial to reduce turnaround time for early diagnosis and treatment by providing the safety and pleasure of patients [26]." Please add own results and discuss them.

Response: Thanks for your valuable advance. We added our results and discussed them.

4. Page 17-18: "Besides, another study also indicated that in vivo aging of red blood cells is associated with increased cellular density, corresponding to increased cell age [28]. These findings may, in part, explain the increased tendency of hemolysis in adults aged more than 40." In my opinion, here, the increasing cell age have been confused with the patient age. It should be reconsidered.

Response: Thanks for your valuable suggestion. We carefully checked these sentences and modified them in the revised manuscript.

5. Page 19: "Our data revealed that the most common suspected causes of the markedly elevated lipemic index were lipid emulsion (Gastrointestinal Surgery) and hemolysis (Neonatology Department), consistent with these previous reports." Here, please give information about the patients treated with lipid emulsion. Otherwise, this statement should be reconsider.

Response: Thanks for your valuable suggestion. We carefully checked the electronic medical records of patients hospitalized in Gastrointestinal Surgery and modified them in the revised manuscript.

6. Page 20: "creatine" or "creatinine"?

Response: I am very sorry for these mistakes. We carefully checked the whole manuscript to avoid any linguistic or spelling errors. 

7. The "hemolysis in vitro" should be changed to "in vitro hemolysis" thoroughly the manuscript.

Response: We changed "hemolysis in vitro" to "in vitro hemolysis" according to your suggestion.

8. Discussion have been written as if a book chapter. It should be shortened to the purpose of the study.

Response: Thanks for your valuable suggestion. We rewrote the discussion in the revised manuscript.

9. The style of the references should be changed according to the requirements of the journal.

Response: Thanks for your suggestion. The style of the references has been changed according to the requirements of the journal.

Reviewer #2: 

The incidence rate of hemolysis, lipemia，icterus and influence factors: A single-center retrospective study of 501,612 fasting serum biochemistry specimens

Comment: The title should be revised, there is no need to indicate the sample size. Influencing factors may not fit based on the reasons that authors will find in major comments.

Response: We would like to thank you for the appreciation of our submitted manuscript and thank you once more for taking the time and effort to provide these very constructive and insightful suggestions. According to your suggestion, we revised the whole manuscript, revised the title, and rewrote the whole manuscript. 

Major revisions: About Study design: 

Authors should emphasize that this a descriptive study in their methodology . The authors indicated that the objective was to evaluate and link pre analytical factors like diet, drugs, collecting, handling, transportation, and preparing specimens. Yet, in their results, they are not interpreted and related to the HIL. Again, there are no methodology that highlight or show how these factors were collected.

• So, authors should refine their topic as well as their objective and as well as their research question.

Response: We modified the whole manuscript, emphasized that this is a descriptive study in methodology, and refined the topic, objective, and research questions as you suggested.

• Authors should refine their study and emphasize on gender and department according to their results presentation and these the tangible data collected according to their retrospective studies

Response: Thanks for your valuable suggestion. We refined the study and emphasized gender and department according to the actual data collected.

• More of the findings, should be discussed generally and maybe recommended

In discussion part:

Authors should discuss specifically on the effects of the HIL for the lab results according to each parameter. For eg. physiological lipemia may not a wide number of tests being requested. Hemolysis may affect only hematology and maybe Potassium ion results but not others.

Response: We rewrote the discussion in the revised manuscript.

In conclusion part: Authors said they studied influence of factors of HIL, I suggest they should remove these factors as they were not investigated, rather they assumed theoretically as seen in their discussion.

Response: According to your suggestion, we removed not investigated factors.

Authors should rather emphasize on incidence rate as the topic highlights.

Response: Many thanks for your insightful suggestion. We also emphasized the incidence rate as the topic highlights.

---

## [Decision Letter · Decision Letter 1]

5 Jan 2022

The incidence rate and influence factors of hemolysis, lipemia , icterus in fasting serum biochemistry specimens

PONE-D-21-18605R1

Dear Dr. Tian,

We’re pleased to inform you that your manuscript has been judged scientifically suitable for publication and will be formally accepted for publication once it meets all outstanding technical requirements.

Kind regards,

Colin Johnson, Ph.D.

Academic Editor

PLOS ONE

Additional Editor Comments (optional):

Reviewers' comments:

Reviewer's Responses to Questions

**Comments to the Author**

1. If the authors have adequately addressed your comments raised in a previous round of review and you feel that this manuscript is now acceptable for publication, you may indicate that here to bypass the “Comments to the Author” section, enter your conflict of interest statement in the “Confidential to Editor” section, and submit your "Accept" recommendation.

Reviewer #1: All comments have been addressed

Reviewer #2: All comments have been addressed

2. Is the manuscript technically sound, and do the data support the conclusions?

Reviewer #1: Yes

Reviewer #2: Yes

3. Has the statistical analysis been performed appropriately and rigorously? 

Reviewer #1: Yes

Reviewer #2: Yes

4. Have the authors made all data underlying the findings in their manuscript fully available?

Reviewer #1: Yes

Reviewer #2: Yes

5. Is the manuscript presented in an intelligible fashion and written in standard English?

Reviewer #1: Yes

Reviewer #2: Yes

6. Review Comments to the Author

Reviewer #1: Thank you for the revision of the manuscript. Authors have properly replied to all my comments. I have no further comments.

Reviewer #2: The authors have revised the title, objectives and methodologies. The authors also revised their discussion against the results found. The authors have addressed comments.

7. PLOS authors have the option to publish the peer review history of their article (what does this mean?). If published, this will include your full peer review and any attached files.

Reviewer #1: **Yes: **Şerif Ercan

Reviewer #2: **Yes: **Jean Baptiste Niyibizi

---

## [Editor Report · Acceptance letter]

7 Jan 2022

PONE-D-21-18605R1 

The incidence rate and influence factors of hemolysis, lipemia, icterus in fasting serum biochemistry specimens 

Dear Dr. Tian:

I'm pleased to inform you that your manuscript has been deemed suitable for publication in PLOS ONE. Congratulations! Your manuscript is now with our production department. 

Kind regards, 

on behalf of

Dr. Colin Johnson 

Academic Editor

PLOS ONE